# Optimizing Operating Points for High Performance Lesion Detection and Segmentation Using Lesion Size Reweighting

**Brennan Nichyporuk**[1]                                                   brennan.nichyporuk@mail.mcgill.ca
**Justin Szeto**[1]                                                            justin.szeto@mail.mcgill.ca
**Douglas L. Arnold**[2]                                                       douglas.arnold@mcgill.ca
**Tal Arbel**[1]                                                                   arbel@cim.mcgill.ca

[1] *Centre for Intelligent Machines, McGill University, MILA, Canada*

[2] *Montreal Neurological Institute, McGill University, Montreal, Canada*

## Abstract

There are many clinical contexts which require accurate detection and segmentation of all focal pathologies (e.g. lesions, tumours) in patient images. In cases where there are a mix of small and large lesions, standard binary cross entropy loss will result in better segmentation of large lesions at the expense of missing small ones. Adjusting the operating point to accurately detect all lesions generally leads to oversegmentation of large lesions. In this work, we propose a novel reweighing strategy to eliminate this performance gap, increasing small pathology detection performance while maintaining segmentation accuracy. We show that our reweighing strategy vastly outperforms competing strategies based on experiments on a large scale, multi-scanner, multi-center dataset of Multiple Sclerosis patient images.

**Keywords:** lesion, segmentation, CNN

## 1. Introduction

Many clinical contexts require accurate detection and segmentation of multiple lesions of varying sizes within a single patient image, either to diagnose or stage a disease or determine treatment efficacy (Doyle et al., 2017). Methods based on the UNet architecture (Ronneberger et al., 2015) use pixel-wise loss functions to learn the appropriate segmentation output given an input MRI and a target. Although voxel-wise loss functions have proven effective to train models to produce accurate segmentations as measured by voxel-wise metrics such as DICE, they suffer from an inherent bias towards larger lesions that contain more voxels. As a result, voxel-wise loss functions typically miss smaller lesions at operating points that are favorable to voxel-wise metrics such as DICE (Nair et al., 2020). Reducing the detection threshold to an operating point that is more suited for detection is a feasible workaround, but this comes at the cost of over-segmenting larger lesions. Given that the optimal operating point for detection and segmentation are different, simultaneously achieving both objectives is not possible with standard loss functions. Recent research (Shirokikh et al., 2020) suggests that re-weighing the voxels of each lesion in a manner that is inversely proportional to that lesion's size can be an effective way to improve small lesion detection performance. Although this approach directly deals with the size imbalance between multiple lesions, it assigns equal weight to each lesion, which can be problematic in contexts such as cancer and Multiple Sclerosis, where lesions span a wide range of sizes (and can be quite small). In this work, we propose a novel weighing function that is much less

prone to the training instability caused by assigning a high weight to smaller lesions that are typically much more uncertain. Our approach closes the detection/segmentation performance gap, showing that with the right lesion reweighing strategy, high overall simultaneous detection and segmentation accuracies are possible. Through large scale experiments on a large propriety dataset of Multiple Sclerosis patient images, the proposed method outperforms the competing baseline and several other common loss functions.

## 2. Lesion Size Reweighting

We propose a lesion weighing function, where the objective is to have the optimal detection and segmentation operating points converge by assigning more weight to small lesions than would otherwise be assigned by binary cross entropy. Although small lesions can be weighed more, they should still be assigned less weight than larger lesions, which are typically much more certain. Our conjecture is that assigning too much weight to small lesions can produce suboptimal results.

Formally, each lesion $L_j$ is assigned a weight $W_j$ that is a function of the number of voxels $|L_j|$ that comprise that lesion. In practice, weights must be assigned to individual voxels rather than individual lesions, so we also define the voxel weight $w_j$, related to $W_j$ via $w_j = \frac{W_j}{|L_j|}$.

$$W_j = |L_j| + \alpha e^{-\frac{1}{\beta}(|L_j|-1)} \qquad w_j = 1 + \frac{\alpha}{|L_j|}e^{-\frac{1}{\beta}(|L_j|-1)} \tag{1}$$

where $\alpha$ and $\beta$ are hyperparameters such that $\alpha \leq \beta$ to ensure monotonicity in the weight with respect to lesion size. Background (i.e. non lesions) voxels retain a weight of 1.

## 3. Experiments and Results

We train a UNet architecture (Ronneberger et al., 2015) to segment T2 lesions with binary cross entropy (BCE), weighted BCE (WBCE), focal loss (FL) (Lin et al., 2017), BCE with the proposed lesion size reweighting (BCE+LSR), and BCE with inverse weighting (BCE+IW) (Shirokikh et al., 2020). Hyperparameters common to all methods (augmentation, dropout, etc.) were first tuned for our baseline BCE model. We then freeze these hyperparameters for all subsequent experiments, modifying only the loss function and learning rate. Hyperparameters for the proposed BCE+LSR loss function were tuned on a $log_2$ scale ($\alpha = 4$ and $\beta = 4$ performed best in our experiments). Our dataset (train/validation/test), contains 1350/175/175 MRI scans from 575/175/175 subjects obtained over the course of a 2 year clinical trial. The train split contains 1-3 scans per subject, each taken 1 year apart. MRI sequences used include FLAIR, PDW, T2, T1, and gadolinium enhanced T1.

Figure 1 shows the TPR vs FDR curves and compares overall segmentation performance with detection performance for small (3-10 voxels), medium (11-50 voxels), and large (51+ voxels) lesions for the proposed BCE+LSR, as compared to BCE and BCE+IW. In the case of BCE+LSR, the optimal operating points for segmentation and detection (red and blue dots) overlap and the method performs well on both tasks. This is in contrast to BCE, for which the optimal operating points are comparatively far apart, and which shows a degree of over-segmentation at the optimal detection operating points (and under-detection

at the optimal segmentation operating point, particularly for small lesions). WBCE and FL exhibited performance characteristics similar to BCE. For BCE+IW, the distance between the optimal detection and segmentation operating points is even larger, and the method significantly underperforms all others. Given the significant decrease in performance for BCE+IW relative to both BCE and BCE+LSR, further analysis revealed that BCE+IW applied substantial weight to extremely small lesions. Since the lesion weights computed by BCE+IW ranged over several orders of magnitude, training was extremely unstable. On the other hand, using the proposed BCE+LSR, the weights remain in a reasonable range, upper bounded by $1 + \frac{\alpha}{|L_j|}$. Since smaller lesions are considerably more uncertain, using a weighting scheme with a reasonable upper bound prevented training instability.

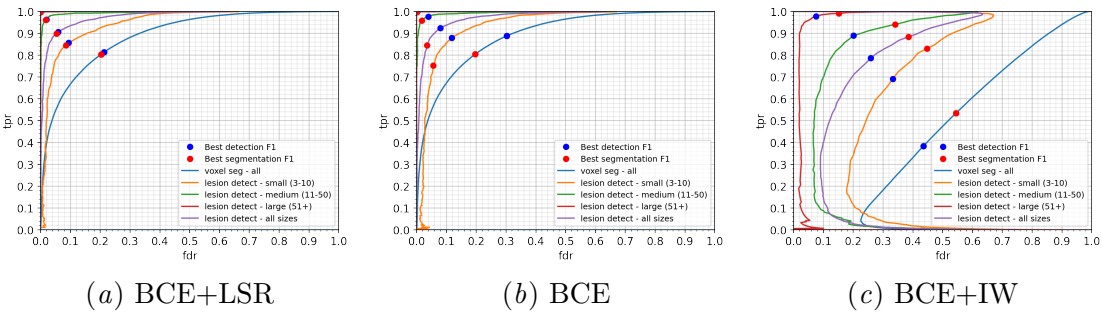

$(a)$ BCE+LSR $\qquad\qquad\qquad$ $(b)$ BCE $\qquad\qquad\qquad$ $(c)$ BCE+IW

Figure 1: TPR vs FDR curves: voxel-level segmentation and lesion-level detection. The best detection F1 operating point (blue dot) is based on the *lesion detect - all sizes* curve. The best segmentation F1 operating point (red dot) is based on the *voxel seg - all* curve. The closer the operating points the better. The operating points overlap for the proposed BCE+LSR (i.e. BCE+LSR achieves the highest simultaneous detection and segmentation F1).

## Acknowledgments

This work was supported by an award from the International Progressive MS Alliance (PA-1603- 08175) and by funding from the Canada CIFAR AI Chairs Program.

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
