# OpenReview forum: "Optimizing Operating Points for High Performance Lesion Detection and Segmentation Using Lesion Size Reweighting"
_MIDL.io/2021/Conference/Short — MIDL 2021 Poster_

### Official Review · Reviewer_2Q3q · 2021-04-23

**Confidence:** 5
**Final Rating:** 4

**Summary:**

Authors present a new "Lesion Size Reweighting" to be included in the loss function when training a semantic segmentation method. The authors highlight the common issue with lesion segmentation: proper detection of small lesions is difficult and the used loss-functions usually focus on larger lesions (easier to segment) than smaller lesions (less impact on e.g. Dice loss). Their proposed lesion weighting function addresses this, by assigning more 'importance' to small objects in the loss function (taking into account their size).

**Strengths:**

Authors address an important point and provide a good solution for it. Detection of small lesions is very hard, as demonstrated in various MICCAI challenges over the past years. The presented formula is simple and efficient; and avoid the effect of "importance explosion" with the earlier published inverse weighting approach.

**Weaknesses:**

There is no information on the task used to evaluation this method, besides that it is a "large propriety dataset of Multiple Sclerosis patient images". Some more information on the data set (number of subjects, type of images, modality) should be provided. Or a reference to a prior publication on this dataset.

Figure 1: operating points for detection and segmentation come closer. Authors should add AUC values as well. It is not clear whether overall detection/segmentation performance for small lesions also improves (does the orange line move up-left)?

Authors might consider applying this task on a public dataset. E.g. the WMH Segmentation Challenge of MICCAI 2017, where it is noted that small lesion detection is an issue.

An overall issue (not per se related to this work) is that many authors use segmentation methods / metrics / loss-functions to develop a detection method. Detection and segmentation require different approaches, but very often segmentation methods (e.g. nnU-Net) are deployed to perform a detection task. The poor performance on small lesions can be improved by making small tweaks (e.g. as proposed here: modifying the loss), but the true solution might be a totally different approach: a dedicated detection method.

**Deanonymize Review:**

no

**Justification Of The Rating:**

Clear and concise paper. To the point. Nice results.

Filler text to meet the character count: Lorem ipsum dolor sit amet, consectetur adipiscing elit. Maecenas cursus sapien id risus convallis mollis. Donec sit amet mattis lectus. Praesent faucibus vulputate justo, ac condimentum diam gravida.

**Paper Type:**

methodological development

**Special Issue:**

yes

---

### Official Review · Reviewer_2vTy · 2021-04-26

**Confidence:** 3
**Final Rating:** 2

**Summary:**

In this paper, the authors suggest a weighting strategy for the loss, to better reconcile the detection and segmentation performances of a DNN (here, a Unet).
The paper is overall well written, and the proposed strategy is easy to understand, and to implement. It consists in assigning a weight to each pixel, that is a function of the lesion size.
The method is compared to a standard baseline (simple BCE), but also a more recently published strategy (BCE + IW).
The code has been made partially available.

**Strengths:**

- The paper is overall well written and easy to understand.
- Although the suggested reweighting is quite simple, it could be an interesting way to handle the problem, and its simplicity could lead to a quick implementation by other research groups.
- The results look promising, and although the experimental setup could be better described, there is a comparison with other methods, serving as an acceptable baseline.


**Weaknesses:**

There are still some points that should be better described when it comes to the experimental setup & results (Third section):
- Even though I understand that the data could not be made available (and that there is a strict 3-page limit), I would have liked to see a quick description: how many patients were there in the database, on which data is the evaluation performed, etc. I think it is important in view of publishing reproducible work;
- Figure 1 is a great way to visualize the gap between the operating points, but I would have also liked to see the actual detection and segmentation performance (especially to compare between BCE & BCE + LSR). On the curves of Figure 1, it seems that the performance of the suggested method might be just below the performance when training only with the BCE (maybe it is just a visual impression...). Is there a tradeoff between performance and having closer operating points ?
- WBCE and FL are mentioned at the beginning of section 3, but I see no evaluation further in the paper... Was it because it was performing less well ?
- How did you choose the values for the hyperparameters $\alpha$ and $\beta$ ?


**Deanonymize Review:**

no

**Detailed Comments:**

Regarding the second point raised in the weaknesses section, having a table summarizing the detection & segmentation performances, as well as the actual values of the thresholds could be a good way to present it.


**Justification Of The Rating:**

Even though I found the paper interesting, I am giving it a weak reject.
The general idea is clear, the problem is well explained, but I think some clarifications are required, especially on the experimental setup, and that I listed in the weaknesses section - especially the first two points. I think they are important questions in order to be able to better judge:
- the reproduciblity of the work;
- the quantitative performance of the method.

However, I think that if the authors manage to adress my comments (and I think  it can be done without having to re-run any experiment), it could make an interesting contribution as a short paper.

**Paper Type:**

both

**Special Issue:**

no

---

### Meta-Review · Program_Chairs · 2021-05-11

**Recommendation:** Accept (Poster)
**Confidence:** 5

**Metareview:**

Two reviewers have different opinions about this work. In genereal, this paper is concise and the results are promising. I maintain the acceptance for this conference.

---

### Decision · Program_Chairs · 2021-05-11

Accept (Poster)